# Influence of Lubricant Environment on Machined Surface Quality in Single-Point Diamond Turning of Ferrous Metal

**DOI:** 10.3390/mi12091110

**Published:** 2021-09-15

**Authors:** Menghua Zhou, Jianpeng Wang, Guoqing Zhang

**Affiliations:** 1Shenzhen Key Laboratory of High Performance Nontraditional Manufacturing, College of Mechatronics and Control Engineering, Shenzhen University, Nan-hai Ave 3688, Shenzhen 518060, China; 1810293032@emial.szu.edu.cn (M.Z.); 1900291014@email.szu.edu.cn (J.W.); 2Levoit Structure Research and Development Department, Vesync Company Limited, Zhongshanyuan Ave, Shenzhen 518060, China; 3Guangdong Key Laboratory of Electromagnetic Control and Intelligent Robots, College of Mechatronics and Control Engineering, Shenzhen University, Nan-hai Ave 3688, Shenzhen 518060, China

**Keywords:** diamond tool, single-point diamond turning, lubricant, ferrous metal

## Abstract

In the field of single-point diamond turning (SPDT), machining ferrous metal is an important research topic with promising application. For SPDT of ferrous metal, the influence of lubricant on the workpiece surface morphology remains to be studied. In this study, three lubricant machining environments were selected to carry out specific control experiments. The machined surface morphology and cutting force in different lubricant machining environments were analyzed. The experiment results showed that the lubricant environment will have significant impacts on the quality of the machined surface morphology of ferrous metal. In the environment of minimum quantity lubrication machining (MQLM-oil), better machined surface quality can be obtained than that in ordinary dry machining (ODM) and high-pressure gas machining (HGM). Furthermore, the cutting force captured in the ODM and HGM environment increased with the increase of the cutting depth, while the cutting force in the MQLM-oil environment remained almost unchanged. That indicates MQLM-oil can suppress the formation of hard particles to improve the machining quality.

## 1. Introduction

Ferrous metals are widely applied in industrial production due to their excellent mechanical properties, while can cause catastrophic tool wear in single-point diamond turning (SPDT) of ferrous metal. Furthermore, grinding, polishing, etc. applied in ultraprecision machining of ferrous metal have low machining efficiency which is difficult to meet the current manufacturing needs [1,2]. Therefore, researches related to STDP ferrous metal is necessary. 

Single-crystal diamonds have extremely high hardness and can be ground to produce extremely sharp cutting edges, which can provide important support for controllability in ultraprecision turning [3]. However, when diamond tools are employed to machine ferrous metal, chemical reactions will lead to catastrophic diamond tool wear to lose the ability to machine steel molds for optical components [4,5]. Catastrophic wear is affected by many factors, including the workpiece material, machining parameters, environment etc. [6] Paul et al. suggested that the unpaired d-electrons induced a chemical reaction between Fe and C [7]. Furthermore, high temperature and high pressure in the machining process increased the chemical reaction speed to increase the wear of diamond tools [8]. Compared with the wear in turning nonferrous metals, mechanical wear of diamond tools accounts for a small part [9,10]. In fact, numerous complex chemical reactions lead to the catastrophic chemical wear of diamond tool, whereas the dominant chemical wear is graphitization [11]. In Komanduri’s study, comb-like groove wear was captured, and the groove direction is consistent with the cutting direction [10]. It is hard to have scratch patterns on the diamond surface due to weak hardness of low-carbon steel, therefore they propose a mechanism of graphitization wear of diamond tools which can weaken the diamond hardness [10,11,12]. Thornton and Wilks further studied the graphitization of diamond, and the experimental results showed that the temperature of diamond graphitization is about 1800 K under static vacuum conditions, while that is between 1000~1100 K in the presence of Fe [13]. However, because the process of SPDT ferrous metals is not static contact, Narulkar et al. further researched the graphitization mechanism of diamond by employing molecular dynamics (MD) and showed that iron catalysis plays an important role leading to the graphitization of diamond tools [14]. 

To suppress diamond tool wear in SPDT ferrous metals, many assisted methods have been developed and mainly through the way of improved machining technology, modified diamond tools and workpiece material [15]. At present, ultrasonic vibration-assisted cutting (UVC) is the most effective machining technology to suppress diamond graphitization wear in cutting ferrous metals and can achieve the mirror-level machined surface roughness [16,17]. Furthermore, assisted method employing CO_2_, CO, CH_4_, C_2_H_2_ etc. are unable to enhance effectively diamond tool life, due to the close contact between workpieces and diamond tools [18,19]. Focusing on the tool properties, nanodiamond coating and diamond tools modified by ion implantation have been developed to suppress graphitization of diamond [20,21]. However, diamond tool wear is still serious. Furthermore, assisted method of workpiece nitriding has also been developed and can achieve mirror-level machined surface roughness, which is nonetheless limited on the nitriding layer [22]. More importantly, these assisted methods are subject to their own limitations and rarely applied in industry production. Therefore, the method of improving machined surface quality, especially in-depth insight into physical or chemical mechanisms in SPDT ferrous metals, is still needed. 

Lubricant is wildly applied in machining process as a typically physical cooling and lubrication method [23]. More importantly, lubricant has a significant influence on diamond wear in SPDT ferrous metals [24]. Therefore, researches on the lubricant is one of the most basic topics in SPDT ferrous metals. However, the influence of lubricant on the machined surface quality in SPDT ferrous metals is still unclear. In this study, by single-groove scraping experiments, an in-depth insight into the physical mechanism of the influence of lubricant environments to the machined surface quality is provided, and provides an important reference for the subsequent research on relevant aspects. 

## 2. Experiments

To explore the influence of lubricant on SPDT ferrous metals, this section will focus on the two sets of single-groove scraping experiments, and conduct a detailed and in-depth analysis of the experimental results. Benefitting from high measurement accuracy, white light interferometer is a typical measurement apparatus to capture the surface morphology of workpieces in ultraprecision machining. In this study, the morphology of machined surfaces and grooves was captured by means of the white light interferometer (Contour GT-X, Bruker, Billerica, MA, USA) by vertical scanning interferometry (VSI) mode with no filter. Appling relatively low-hardness die steel AISI 4140 (Shengjili Co., Ltd., Sichuan, China) to carry out specific experiments. The cutting force was captured by an ultra-precision dynamometer (Kistler 9017C, Winterthur, Switzerland). Before the single-groove scraping experiment, the workpieces were rough-machined with a polycrystalline diamond (PCD) tool (Shenzhen Yuhe Diamond Tools Co., Ltd., Shenzhen, China). Three machining environments of minimum-quantity lubrication machining (MQLM-oil), ordinary dry machining (ODM) and high-pressure gas machining (HGM) were selected to rough-machine the workpiece surface. The surface morphology and roughness obtained by the two machining environments of HGM and ODM were similar. The specific rough-machined surface morphology was captured by means of the white light interferometer, as shown in Figure 1. The value of the surface roughness *Sa* (see Figure 1) indicates that MQLM-oil is beneficial to improving the machined quality compared with ODM and HGM.

The single-groove scraping experiments were divided into two parts, employing four natural single-crystal diamond tools (Shenzhen Yuhe Diamond Tools Co., Ltd., Shenzhen, China) T1 with 496.2 μm radius, T2 with 490.9 μm radius, T3 with 492.6 μm and T4 with 483 μm radius. All of the diamond tools have the rake angle of 0°and the clearance angle of 10°. The first part was the 360° single-groove scraping experiment on the rough-machined surface obtained in three machining environments. The theoretical depth of the single groove is 12 μm. The feed of Z-axis was achieved by 2 μm, and the cutting force was captured by the ultraprecision dynamometer, as shown in E-1 to E-3 in Table 1. The second part was also carried on the rough-machined surface obtained in the three machining environments, as shown in Figure 2. The annular groove was divided into six uniform parts (N1–N6). In the experiments, the feed of Z-axis was 2 μm and the scraping angle of a single-groove was 60°; the cutting force was captured by the ultraprecision dynamometer. After the experiments, the white light interferometer was employed to capture the surface morphology of the workpiece, and the cutting edge of the diamond tool was observed with a metallographic microscope and there was minimal wear of diamond tools in the scratching experiments since the scraping time was very short and the scraping speed was very slow compared to turning. Therefore, the wear status of the employed cutting tool could be negligible.

## 3. Results and Discussions

### 3.1. Surface Morphology of Single-Groove

PCD tools were employed to rough-machine the AISI 4140 workpiece with a diameter of 10 mm in the ODM environment, and the surface morphology with a surface roughness of 0.278 μm is shown in Figure 1b. The scraping depth was 2 μm and the total scraping depth was 12 μm. Figure 3a is part of the surface morphology in the HGM, which is similar to the ODM; Figure 3c is part of the surface morphology in the MQLM-oil. Comparing the experimental results, there were pits on the surface of the groove in HGM and ODM environments, while there were almost no pits on the surface of the groove in the MQLM-oil environment. According to statistics on the occurrence frequency and depth of pits, it was found that pits appeared the most and had the deepest depth in the HGM environment and found that the amount of pits in a single-groove in HGM and ODM was much more than that in the MQLM-oil environment, and the depth of pits in the MQLM environment was also smaller compared with the other two machining environments. The appearance of pits resulted in the terrible morphology of the single-groove, as shown in Figure 3b,d. The details of the depth of the pit in three machining environments are shown in Figure 4.

The experiments of E-1, E-2 and E-3 were repeated on the rough-machined surface, which was machined in the MOLM-oil and ODM environments, respectively. The experimental results of rough-machining in the ODM were almost as the same as the experimental results of the rough-machining in the HGM environment, and the experimental results of rough-machining in the MOLM-oil are shown in Figure 4b. There are numerous pits on the single-groove surface in HGM and ODM environments, and the depth of pits was 12 μm to 16 μm; in MQLM-oil environment, few pits were found on the single-groove surface. Compared the experimental results in the HGM rough-machining environment, it was found that in the single-groove scraping experiment on the surface rough-machined by the HGM environment, there were still a few pits in the MQLM-oil environment.

To explore the influence of the rough-machining environment on the subsequent machining, more detailed scraping experiments were carried out. After rough-machined experiments in the HGM environment, the experiments of N-1 to N-6 were carried out (see Table 1) and the experimental results were shown in Figure 5a. In both HGM and ODM environments, the maximum pit depth will not increase or decrease significantly with the increase of the cumulative cutting depth, while in the MQLM-oil environment, the maximum pit depth will reduce significantly with the increase of the cumulative cutting depth. Repeated experiments are carried out for rough-machining workpieces in HGM and MQLM-oil environments. The rough-machining results in ODM environment are similar to that in HGM environment. Figure 5b shows the details of pits in the MQLM-oil environment. Summarizing the data of the surface pits of the rough-machining in the MQLM-oil environment, it was found that there were no obvious pits on the single-groove surface in this environment, and the results obtained in this rough-machining environment of the other environments were almost the same. The depth increased first and then remained almost unchanged.

Therefore, the MQLM-oil environment had a prominent effect on reducing the number of the appearance of pits. It also proved that other environments will cause a significant body of pits in SPDT ferrous metal. The maximum depth of the pits will not increase indefinitely with the increase of the cumulative cutting depth, but will eventually remain between 12 μm and 16 μm.

### 3.2. Cutting Force Analysis of Single-Groove Scraping Experiments

The cutting force in the above-mentioned experiment was captured and analyzed. The magnitude of force when scraping depth of 2 μm was shown in Figure 6. In Figure 6a are shown the scraping experimental carried out in the three environments when the surface of the workpiece was rough-machined in the MQLM-oil environment. The average cutting forces under the three environments were almost same in the first experiment. The cutting force increased in the three environments with the cutting depth increased. More importantly, the increased force in the HGM environment was the most obvious, whereas that in the MQLM-oil environment was not obvious. Moreover, the average cutting force in the two machining environments of MQLM-oil and HGM was almost equal at a depth of scraping 2–8 μm, while the average cutting force in the HGM environment appeared to significantly increase after scarping depth more than 8 μm.

In Figure 6b are shown the scraping experimental results in the three environments when the surface of the workpiece was rough-machined in the HGM environment. The experimental results of ODM were similar to that of the HGM. The average cutting forces under the three cutting environments were almost same in the first experiment. With the increased depth of cutting, the cutting force was increased in the HGM environment. When the cutting depth was 2 μm, the cutting force captured in MQLM-oil rough-machined was only about 0.6 N, while the surface cutting force obtained by others was about 1.1 N. 

Therefore, the surface of rough-machined will affect the results of the initial scraping experiments, whereas the effect will be less as the cumulative cutting depth increases. Furthermore, the surface morphology of the groove captured in different scraping environments is almost the same. Therefore, the rough-machining environment will have a significant influence on the subsequent cutting, whereas it will not continue to affect that and the critical depth is 12 μm.

### 3.3. Physical Mechanism of the Lubricant Influence on Single-Groove Surface Morphology

The existence of hard carbide particles is the dominant physical mechanism of the lubricant influence on single-groove surface morphology. There are numerous hard carbide particles in AISI 4140. As shown in Figure 7, a trace of hard particles of about 2 μm can be found on the machined surface which are two orders of magnitude harder than the workpiece [1,25]. 

Figure 8a is shown a pit area captured by means of the white light interferometer, and the morphology of the pits are along the cutting direction. The generation mechanism of pits is illustrated in Figure 8b. The pits always present a steep surface on one side and a gentle surface on the other side. Obviously, this is caused by hard particles being pulled out by the diamond tool during the cutting process. In the experiments, the position of the diamond tool installed on the tool holder remains fixed. Therefore, the pit within the material cannot removal by plastic caused by the hardness advantage of diamond tools. The mechanism of pits generation can be clearly expressed in Figure 8b. In the process of single-groove scraping, the speed of the diamond tool is low. When it meets soft ferrous metal, the material can be plastically removed to form a single-groove morphology in low cutting speed. However, the hard particles in the ferrous metal cannot be plastically removed when the tool interferes with the hard particles during the cutting process, therefore the hard particles will be pulled out to form pits.

The appearance of hard carbide particles will lead to the fluctuation of cutting force, therefore it is suitable to employ cutting force to characterize the characteristics of pits in the cutting process. Figure 9 shows the corresponding cutting force and the corresponding single-groove morphology in the three environments. In the environment, with the diamond tool cutting the material beginning, there was a sudden change of cutting force. The same phenomenon also appeared in other two machining environments. However, only the cutting force in the MQLM-oil environment remained basically stable during the subsequent processing; while that of HGM and ODM showed large fluctuations. Because the removal of materials in these two types of environments is unstable, the formation process of pits will inevitably lead to continuous fluctuation in the captured cutting force.

Moreover, under the HGM environments, the cutting force fluctuated for a long time corresponding represents the large volume of pits; under the ODM environments, the cutting force fluctuated for a short time corresponding represents the small volume of pits; under the MQIL-oil environments, the cutting force changed smoothly, which means there were no pits. Therefore, cutting forces can be employed to properly characterize the properties of pits during cutting.

### 3.4. Influence of Closed-Loop-Stiffness of the Lathe on Single-Groove Surface Morphology

Another possible reason is that the closed-loop-stiffness of the ultraprecision lathe was not enough caused the spindle crawled during machining. In the experiment, especially the scraping experiment of the single-groove, the diamond tool was stressed by a large force when removing the workpiece material. The force generated in this process will be transmitted to the spindle eventually. As the aerostatic spindle, the stiffness of the spindle is very limited. When the force is large, the spindle is usually prone to crawling and the pulsed force causes an unstable process of material removal.

As shown in Figure 10a, during the working process of the ultra-precision lathe, the interaction force caused by the interference between the tool and the workpiece will be transmitted through the spindle and lathe bed, and finally form a closed-loop force caused the phenomenon of spindle crawling. As shown in Figure 10b, the tool is mainly subjected to two forces in the Z-axis and Y-axis directions during the machining process. The force in the Z-axis direction may cause the crawl of the Z-axis, and the force in the Y-axis direction may cause the crawl of the C-axis. During the relative sliding process between the spindle and the guide rail, when the driving force of the spindle cannot overcome the resistance, the motion will be stopped, and as the driving equipment subjected to the spindle continues to motion, it will eventually overcome the resistance and leap forward. This phenomenon is commonly known as crawling for the spindle.

Obviously, the lubrication effect produced by different lubricant environments is inconsistent, and the equivalent friction coefficient of MQLM-oil is lower than that of the ODM environment. The equivalent friction coefficient is high, which means that the resistance to overcome increases, and the probability of occurrence of the phenomenon of spindle crawling becomes greater. As shown in Figure 10b, under the action of the resultant force of the tool, if the crawling phenomenon occurs, an irregular tool path will be generated, and the crawling of the tool in the Z-axis direction will cause the single-groove surface to sink downward.

As shown in Figure 11, the pits caused by different factors had different forms in single-groove. The pit morphology caused by the removal of hard particles was limited by the size of the hard particles and could not affect the cross-sectional profile of the entire single-groove, as shown in Figure 11a. The pits caused by the crawling phenomenon were caused by the irregular changes of the tool path. The pits must have affected the cross-sectional of the entire single-groove, as shown in Figure 11b.

## 4. Conclusions

In this study, the influence of the lubricant on machined surface morphology was studied. Three experiments of HGM, MQLM-oil, and ODM were carried out and the cutting force was analyzed. To conclude, we found:(1)The lubrication and cooling method by applying lubricant have an important influence on the surface morphology in SPDT ferrous metals. In the MQLM-oil environment, better surface quality can be obtained than that in the HGM and ODM environments.(2)The existence of hard carbide particles is the main reason for the physical mechanism of pit generation in scraping ferrous metal.(3)Cutting force can be employed to characterize the properties of pits during cutting, the frequency and period of large-scale fluctuation of the cutting force represent the number and size of pits respectively.(4)The closed-loop-stiffness of the ultra-precision lathe is low, resulting in discontinuous and large-area pits on the surface of the ferrous metal.

## Figures and Tables

**Figure 1 micromachines-12-01110-f001:**
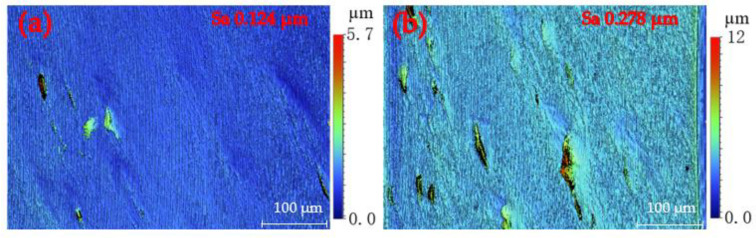
Machined surface morphology captured by means of the white light interferometer in (**a**) minimum quantity lubrication machining (MQLM-oil) and (**b**) high-pressure gas machining (HGM) environments.

**Figure 2 micromachines-12-01110-f002:**
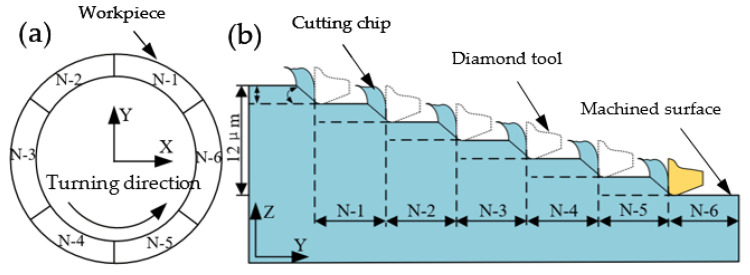
Diagram of (**a**) single-groove scraping experiments on the workpiece and (**b**) cutting depth changed from N-1 to N-6.

**Figure 3 micromachines-12-01110-f003:**
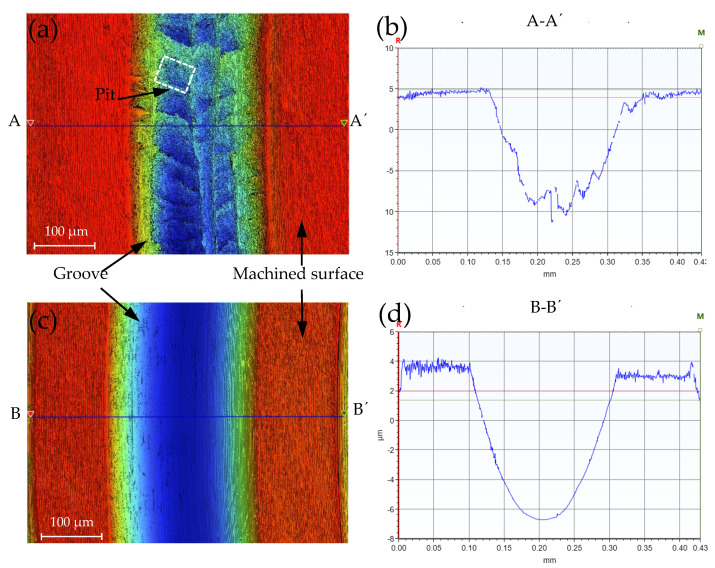
Single-groove scraping morphology captured by means of the white light interferometer in (**a**) HGM environment and (**c**) MQLM-oil environment; (**b**) the cross section curve of A-A’ and (**d**) the cross section curve of B-B’.

**Figure 4 micromachines-12-01110-f004:**
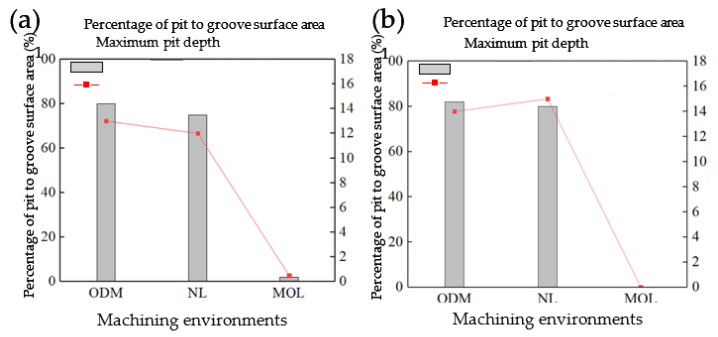
The statistical data of the pits in the groove scraping experiments of E-1, E-2 and E-3 in rough-machined surface machined in (**a**) HGM and (**b**) MQLM-oil environments.

**Figure 5 micromachines-12-01110-f005:**
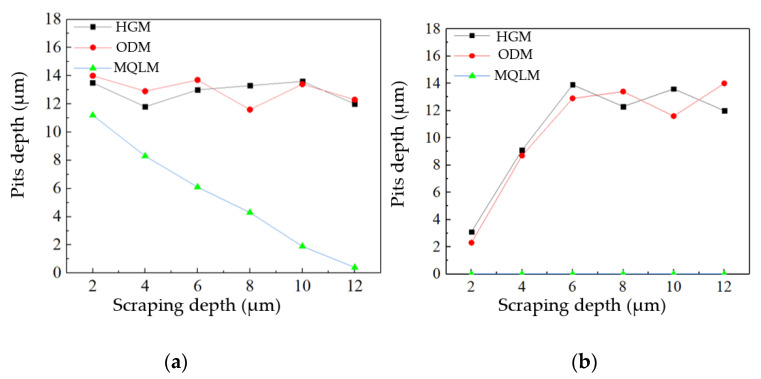
Relation of the scraping depth and the depth of the pits in the rough-machined surface of (**a**) HGM environment and (**b**) MQLM-oil environment.

**Figure 6 micromachines-12-01110-f006:**
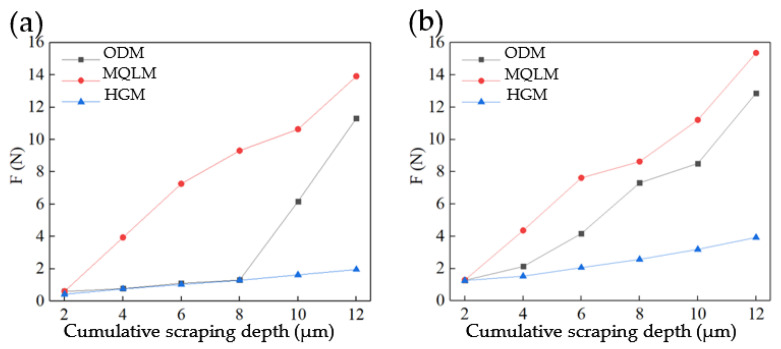
The average force of single-groove scraping experiments of E-1 (MQLM-oil), E-2 (HGM) and E-3 (ordinary dry machining (ODM)) on the rough-machined surface of (**a**) MQLM-oil environment and (**b**) HGM environment.

**Figure 7 micromachines-12-01110-f007:**
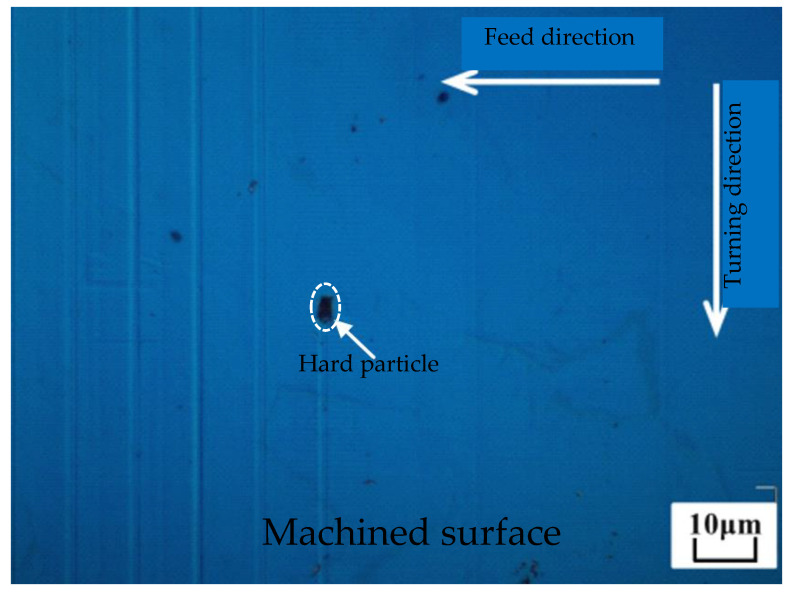
Hard particles on the machined surface in SPDT ferrous metals.

**Figure 8 micromachines-12-01110-f008:**
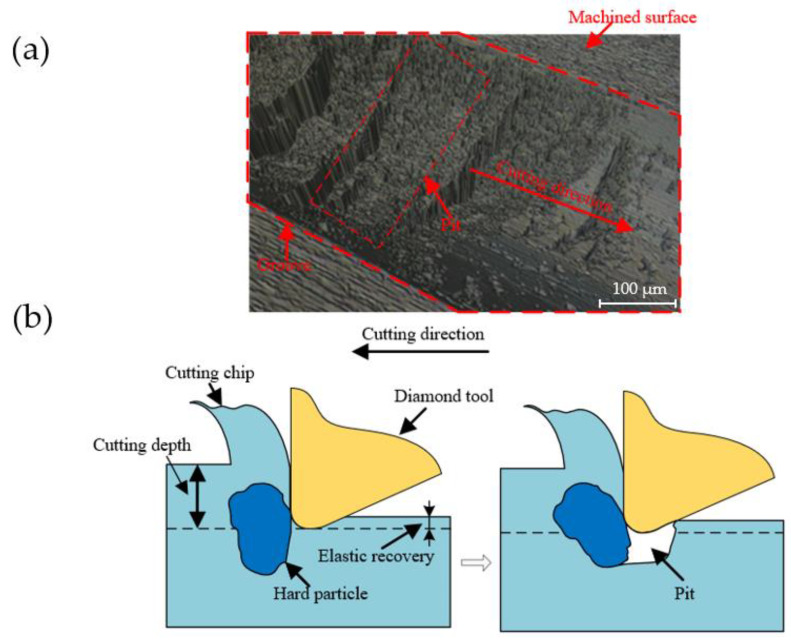
(**a**) Enlarged surface of a pit area on the groove surface and (**b**) the illustration of the pit generation mechanism on the ferrous metal surface in SPDT.

**Figure 9 micromachines-12-01110-f009:**
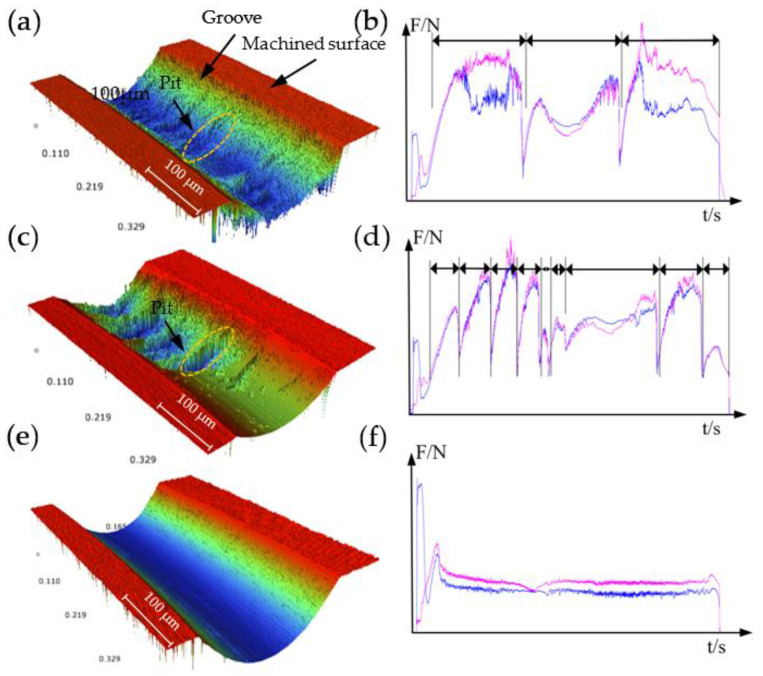
Groove surface morphology and cutting force in (**a**,**b**) HGM environment, (**c**,**d**) ODM environment and (**e**,**f**) MQLM-oil environment.

**Figure 10 micromachines-12-01110-f010:**
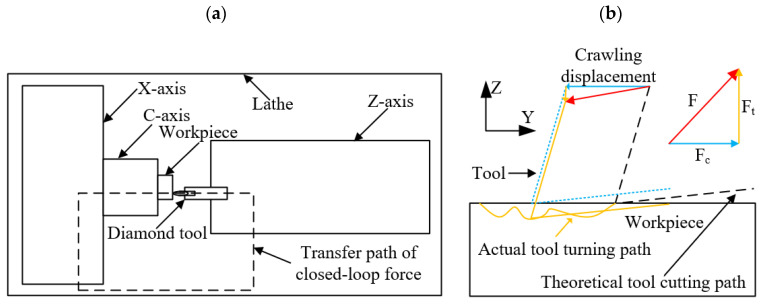
(**a**) Transfer path of closed-loop force on the lathe and (**b**) transfer direction of the force.

**Figure 11 micromachines-12-01110-f011:**
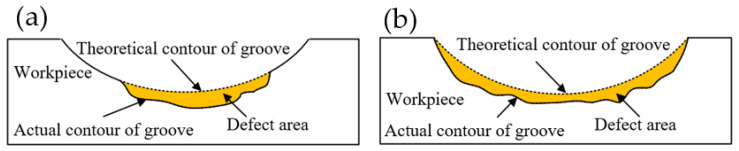
Different types of pit on the groove surface of (**a**) hard particles caused the defect area and (**b**) crawling caused the defect area.

**Table 1 micromachines-12-01110-t001:** Detailed parameters in the single-groove scraping experiments.

Group Number	Rough-Machining Environment	Tool Number	Turning Environment	Single Scraping Depth ∗ Angle	Total Scraping Depth
E-1	OHGM/ODM/MQLM-oil	T1	MQLM-oil	2 μm ∗ 360°	12 μm
E-2	HGM/ODM/MQLM-oil	T2	HGM	2 μm ∗ 360°	12 μm
E-3	HGM/ODM/MQLM-oil	T3	ODM	2 μm ∗ 360°	12 μm
N-1	HGM/ODM/MQLM-oil	T4	MQLM-oil	2 μm ∗ 360°	2 μm
N-2	HGM/ODM/MQLM-oil	T4	MQLM-oil	2 μm ∗ 300°	4 μm
N-3	HGM/ODM/MQLM-oil	T4	MQLM-oil	2 μm ∗ 240°	6 μm
N-4	HGM/ODM/MQLM-oil	T4	MQLM-oil	2 μm ∗ 180°	8 μm
N-5	HGM/ODM/MQLM-oil	T4	MQLM-oil	2 μm ∗ 120°	10 μm
N-6	HGM/ODM/MQLM-oil	T4	MQLM-oil	2 μm ∗ 60°	12 μm

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
