# Peer review of "Influence of Lubricant Environment on Machined Surface Quality in Single-Point Diamond Turning of Ferrous Metal"

_micromachines, 2021, doi:10.3390/mi12091110_

Round 1

Reviewer 1 Report

The manuscript presented an experimental study on the lubricant environment effect on the machined surface quality in ultra-precision diamond turning ferrous metal. The topic is interesting. In the manuscript, the pits left on the machined surface has been fully discussed for the lubricant environment effect. An instruction has been proposed for the improvement of the machined surface quality in ultra-precision diamond turning ferrous metal. The results are significant. It can be accepted for the publication in the journal after minor revision:

  • Add some recent papers published in the journal to better follow its scope.
  • The status of the employed cutting tool in turning should be clearly presented.
  • The abbreviations should be used as less as possible.
  • The scale is missing in some figures, like Fig. 1, Fig. 5, and Fig. 6.
  • The manuscript should be proofread.

Reviewer 2 Report

The main objective of the article was to identify the influence of the lubricant on machined surface morphology.  The paper was well written and developed and could be considered for publication in Micromachines (upon Editor’s approval) provided that some improvements are included in a revised manuscript as follows: The paper is predominantly experimental and the results and discussions of the paper should be organized by planning of the experiments. The physical mechanism must be discussed in a better way  in link the mechanism of the influence of the lubricant on machined surface morphology was studied. The equipment used to perform the measurements should also be presented more clearly. Figures illustrating the morphology of the surface should be made explicit by the presentation and characteristics of the experimental apparatus.

Round 2

Reviewer 2 Report

The authors made the required corrections.